# Novel Cost-Effective Microfluidic Chip Based on Hybrid Fabrication and Its Comprehensive Characterization

**DOI:** 10.3390/s19071719

**Published:** 2019-04-10

**Authors:** Sanja P. Kojic, Goran M. Stojanovic, Vasa Radonic

**Affiliations:** 1Faculty of Technical Sciences, University of Novi Sad, Trg Dositeja Obradovica 6, 21000 Novi Sad, Serbia; sanjakojic@uns.ac.rs (S.P.K.); sgoran@uns.ac.rs (G.M.S.); 2Institute Biosense, University of Novi Sad, Dr Zorana Djindjica 1, 21000 Novi Sad, Serbia

**Keywords:** microfluidic chip, hybrid fabrication, Ceram Tape, PVC foil, xurography

## Abstract

Microfluidics, one of the most attractive and fastest developed areas of modern science and technology, has found a number of applications in medicine, biology and chemistry. To address advanced designing challenges of the microfluidic devices, the research is mainly focused on development of efficient, low-cost and rapid fabrication technology with the wide range of applications. For the first time, this paper presents fabrication of microfluidic chips using hybrid fabrication technology—a grouping of the PVC (polyvinyl chloride) foils and the LTCC (Low Temperature Co-fired Ceramics) Ceram Tape using a combination of a cost-effective xurography technique and a laser micromachining process. Optical and dielectric properties were determined for the fabricated microfluidic chips. A mechanical characterization of the Ceram Tape, as a middle layer in its non-baked condition, has been performed and Young’s modulus and hardness were determined. The obtained results confirm a good potential of the proposed technology for rapid fabrication of low-cost microfluidic chips with high reliability and reproducibility. The conducted microfluidic tests demonstrated that presented microfluidic chips can resist 3000 times higher flow rates than the chips manufactured using standard xurography technique.

## 1. Introduction

Micro and nano-fluidics represent the most progressive areas of modern science and technology, due to a wide range of applications in many fields of our life such as medicine, biology, chemistry, engineering or environmental protection [1]. More precisely, enabling control of fluids on micro-scale, microfluidics chips or lab-on-chip devices can be used in numerous practical applications such as: micromixers, micropumps, droplet generation, biosensors, optical detection systems, and many others. During previous years, various technologies were used for the fabrication of microfluidic devices. To mention some of them, most frequently applied technologies are: PDMS (PolyDiMethylSiloxane), LTCC (Low Temperature Co-fired Ceramic), 3D printing, or xurographic technique. PDMS polymer has many advantages for fabrication of microfluidic chip such as biocompatibility, optical transparency (in the range 240–1100 nm) and mechanical flexibility and stretchability [2]. These properties have opened new fields of application for PDMS-based microfluidic chips for creation of organs-on-chips [3], or point-of-care diagnostics [4]. However, for microfluidic devices, PDMS process requires non-trivial lithography method, for design optimization it is necessary to repeat the complete fabrication flow [5], and the manufacturing of circular channels is a very challenging task [6]. To solve mentioned drawbacks, authors suggested some unconventional techniques, for instance, using lubricant-infused mould [7], combining PDMS membrane with SU-8 and quartz [8], and developing materials with better performances than PDMS [9].

Microfluidic devices can be fabricated also in ceramic-based Low-Temperature Co-fired Ceramic (LTCC) technology, thanks to the possibility to create the complex multilayer structures [10]. For the manufacturing of a microfluidic chip in LTCC process, LTCC tapes are usually used from various producers (e.g., CeramTec, DuPont, Heraeus, ESL) [11], laser for formation of microchannels in desired geometry, and after that lamination and sintering (in a furnace, using carefully selected thermal profile). Before firing, non-baked LTCC tapes are mechanically flexible and numerous geometrical shapes can be cut [12]. Different shapes of laser-micromachined channels were reported such as serpentine, meander, etc. [13,14]. An important advantage of the LTCC process is the possibility to separately test every layer of multilayered structure [15]. LTCC-based microfluidic chips have chemical and temperature stability, very good mechanical properties and the possibility to be combined with structures and components from other technologies [16]. For some applications, drawback of LTCC technology is non-transparency, thus it is necessary to perform bonding of LTCC structure with other transparent materials, such as PDMS [17,18] or glass [19]. Furthermore, a disadvantage of this technology is changing shape and dimensions of microchannels (and sometimes occlusion) during the lamination or firing process [20].

The next interesting technique for microfluidic devices manufacturing is the 3D printing process, through applying additive manufacturing. This process attracted significant attention in previous periods due to the possibility to create complex shapes, quickly and cost-effectively, usually supported by thermoplastic materials, such as acrylonitrile butadiene styrene (ABS) and polylactic acid (PLA) [21]. The printing principle is uninterrupted layer-by-layer, and the final structure is distortion and delamination-free [22]. There are several low-cost printers on the market. The FDM, Polyjet, and DLP-SLA printing techniques for microfluidic chips were compared in [23]. However, limitations of this process are low resolution of the fabricated channels and materials, which are usually used are not optically transparent (which is very important for various microfluidic applications). The channel widths which can be achieved are approximately 150 μm [24], but in most cases the leak-proof channels are wider than 800 μm [25]. To improve performances of 3D printed microfluidic devices, different solutions were proposed such as: using multilaterals 3D printing [26], combination of 3D printing (3D moulds) and PDMS [27,28,29]. Moreover, xurography can be used as a rapid prototyping technique for the rapid manufacturing of cheap microfluidic chips [30], but they have disadvantages in uneven edges of the microchannels. Research groups in the field of microfluidics usually used one fabrication process for the creation of the microfluidics chips, thus there is a lack of studies which report chip design using a combination of different processes. 

In this work, we present a solution for the rapid prototyping of microfluidic chips using a combination of laser micromachining and xurographic technique, without using expensive clean-room facility. The proposed chip combines two materials, PVC and Green Tape, both with relatively good bio-compatible characteristics. PVC is a widely used thermoplastic material in the medical device industry and it is dominantly used for the storage of fluids, dialysis solutions, blood, and blood products [31,32]. PVC has the ability to accept or transmit a variety of fluid without any significant changes in composition or properties [33]. It is characterized by good biocompatibility, which can be further increased by appropriate surface modification [34]. 

Ceram Tape is a LTCC glass ceramic base material composed of an anorthite glass (calcium aluminosilicate) with ceramic filler. This green tape is appropriate for manufacturing fine structures for electronic and microfluidic applications intended to work in a harsh environment. The bio-compatibility of Ceram Tape GC for cell grown has been confirmed in [35]. To investigate the bio–compatibility, proliferation, viability, and adherence of cells on Ceram Tape GC, HeLa (human, cervix epithelial) cells have been grown on sintered tapes using standard test procedures, and the results confirm that Ceram Tape GC material does not affect the viability and adherence of the cells. On the other hand, a number of short- or long-time exposure of the microfluidic chip to different fluids have been accomplished to approve bio-compatibility of the Ceram Tape GC material [36,37].

The complete optical, dielectric, temperature and mechanical characterizations of the fabricated chips were performed. To demonstrate the applicability of the proposed technology concept, a resonant microwave sensor for the monitoring of the fluid properties flowing in the microchannel was designed and tested. Moreover, a multilayered 3D microfluidic chip for pollen filtration that combine a mixer, filtration unit, and serpentine was also fabricated and tested.

## 2. Materials and Methods

### 2.1. Materials and Tools for Microfluidic Chip Fabrication

#### 2.1.1. Materials

For the fabrication of the proposed microfluidic chips PVC foil—A4 hot lamination foil (MBL^®^ 80MIC and MBL^®^ 125MIC, Belgrade, Serbia) with the thickness of 80 μm and 125 μm and Ceram Tape (Ceram Tape GC, CERAMTEC GmbH^®^, Plochingen, Germany) were used. Filter papers (Whatman^®^ 1441-150, Clifton, NJ, USA) with grades of 20–25 µm are used for filtration pads. Deionised water (Grade 2 by ISO 3696 (1987)), blue water based food colouring (Aroma 1990^®^, Belgrade, Serbia), Isopropyl (99.7%, Sigma-Aldrich, Saint Louis, MO, USA), Methanol (99.8%, Sigma-Aldrich), sunflower oil (Vital, Vrbas, Serbia) and pollen (isolated *typha* pollen particles) are used for microfluidic experiments.

#### 2.1.2. Equipment and Small Tools

Plotter cutter (CE6000-60 PLUS^®^, Graphtec America, Inc., Irvine, CA, USA) with the 45° cutting blade (CB09U) and the cutting mat (12” Silhouette Cameo Cutting Mat, Sacramento, USA) were used for carving inlets, outlets and edges of PVC layers for microfluidic chips while Ceram Tapes and filter paper were cut out with laser (Rofin-Sinar Power Line D-100, Germany). Bondage between PVC and Ceram Tape are performed through lamination with A4 card laminator (FG320, Minoan Binding Laminating, Belgrade, Serbia). Bondage between Ceram Tape layers was made with uniaxial press (Carver^®^ 3895CEB, Wabash, USA).

### 2.2. Methods for Microfluidic Chip Fabrication

The proposed microfluidic chip consists of three layers, as shown in Figure 1. Top and bottom layers were realized using PVC foils, while the Ceram Tape was used for the middle layer. Fabrication of the chip was relized through of several steps. In the first step, laser cutting of the middle chip layer in the Ceram Tape, as for standard preparation of Ceram Tape for LTCC technology was performed. For the multilayered chip, that comprises of several Ceram Tape layers bonding with uniaxial press has been done in second step. Plotter cutting of PVC layers, as for standard xurography technique, was used for cutting the inlets and outlet of the microfluidic channels. In the final step, lamination of the cut layers as for standard xurography technique (laminated first: Layer 1 and 2, and then Layer 3) has been accomplished ordered as in Figure 1a. Figure 1 also shows photographs of different designs (simple channel, short and long serpentine) cut in Ceram Tape, using laser. In all chips, microfluidic channels have been realized with the width of 200 µm, while the inlet and outlet holes have been realized with the diameter of 2 mm. The exact fabrication parameters used for manufacturing chips, presented in this paper, are listed in Table 1.

### 2.3. Characterization Techniques

The following instruments were used for characterization and testing of the fabricated microfluidic chips. Optical characterization has been performed using UV-visible spectrophotometer INESA L6S in the wavelength range between 300 nm and 1000 nm. Dielectric characterization has been performed using Agilent VNA E5071C in the frequency range between 500 MHz and 8.5 GHz. SEM for micrographs of fabricated channels was done with the TM3030 Hitachi, Japan, while the Profiler Huwitz Panasis with bioimaging software for 3D profile of microfluidic channels was used for profiler analysis. Mechanical characterization using nanoindentation method was performed using Agilent Nanoindenter G200. Temperature exposure is performed in Memmert oven UN20, Germany. For microfluidic measurements syringe pump model NE 4000 Multi Pulser, KF Technology, Italy, and camera Digital USB Microscope, S02 (1000×), China for recording microfluidic flow in the chip were used.

## 3. Results

### 3.1. Optical Transmission

Light transmittance of the proposed microfluidic chips was measured in the wavelength range from 300 nm to 1000 nm. Four different configurations have been analysed in which the foil was placed on one or both sides on Ceram Tape for two different foil thicknesses: 80 µm and 125 µm. Figure 2 shows the optical transmission as a function of the wavelength for four different configurations. The proposed microfluidic chips show good optical properties with transmittance better than 80%, while the wavelength dependence shows a small variation above wavelength of 340 nm. Microfluidic chips fabricated using PVC foils have slightly lower percentage of optical transmittance comparing with PDMS, but higher than all transparent materials obtained with 3D printing process.

### 3.2. Dielectric Properties

The effective permittivity of the proposed multi-layered configuration has been determined using a phase shift method, i.e., measurement of the phase delay of the sinusoidal signal that propagate along the transmission line [38,39]. For this purpose, the microstrip line was placed on the multi-layered substrate made of the combination of Ceram Tape and foil, as depicted in Figure 3. The conductive layers, namely microstrip line and ground layer, have been realized using sticky aluminum tape. Based on the four set of measurements of two lines with different lengths and two thicknesses of the foil, the effective permittivity of the combination of the inhomogeneous dielectric substrate has been calculated using an equation for effective dielectric permittivity of the multi-layered substrate [38]. The phase and amplitude characteristics of the microstrip line has been determined using Agilent VNA E5071C in the frequency range between 500 MHz and 8.5 GHz. The Figure 4 shows the measurement setup, whereas Figure 5a depicts the extracted values of the effective permittivity of the multi-layered configuration and effective permittivity of each layer. The dielectric losses shown in Figure 5b were calculated based on amplitude of the propagation signal [39].

### 3.3. SEM and Profiler Analysis

For determination of the minimal functional channel width, we made a simple chip structure with six different widths of the channel equal to 50 μm, 100 μm, 200 μm, 300 μm, 500 μm and 1000 μm with one inlet, outlet and observation field per channel. Figure 6 shows one of these chips in the SEM.

Verification and quality of cuts was observed using different magnification rate from 80 to 1200 times, by means of SEM. Figure 7 presents laser cut channels with different widths. It can be confirmed that the measured width of narrower channels is larger than the predefined value. For channels of 100 μm and wider, the laser beam can be adjusted so the exact width can be achieved, while for channel with the width of 50 μm laser has only two passes and it is not possible to further improve its dimensions. Nevertheless, from four chips that were made (two laminated with 80 μm and 2 with 125 μm foil) all channels were fully functional.

Figure 8 shows SEM micrograph of edges of the laser cut Ceram Tape at high magnification. As it can be perceived, edges are quite flat even on magnification of 1200 times, with the irregularities smaller than 10 μm.

On the profiler, we checked the thickness of the Ceram Tape layer. As it can be seen in Figure 9, the thickness of the layer is around 280 μm (in the fabrication data sheet it is about 300 μm). Roughness of the laser cut is not greater than roughness of the surface of the material.

Beside channels, microfluidic chambers with rectangular and circular shapes were made. The width of the rectangle chamber and the diameter of the circle one were set to: I = 1 mm, II = 2 mm, III = 3 mm, IV = 4 mm, and V = 5 mm, while the length of the rectangle chambers is set to 10 mm. As it can be seen in Figure 10, for rectangle chambers, the maximum obtained width for lamination with 80 μm foils on both sides was 3 mm, while for 125 μm foils was 2 mm. Connection between the top and bottom layers made of PVC occurs for larger chamber widths as a consequence of large area of lamination and deformation of PVC foil caused by temperature. Maximal allowed diameter of circle chambers is 4 mm, while on 5 mm diameter midpoint of the chamber is impaired. All chips for microfluidic chambers testing were laminated at 150 °C.

### 3.4. Mechanical Characterization

Mechanical characterisation was conducted on middle layer, i.e., Ceram Tape layer using Agilent Nanoindenter G200. Measurement parameters were set to: max. load 10 mN, peak hold time 10 s, time to load 15 s, allowable drift rate 0.2 nm/s and Poisson ratio 0.32. Figure 11 displays Load vs. displacement curves of 100 measurement points fairly distributed on the sample in the form of 10 × 10 matrix, with 150 μm distance between measurement points. Measured Young’s modulus and Hardness were 175 GPa (st. dev. 0.175) and 0.005 GPa (st. dev. 0.005), respectively. Obtained results confirm that these two materials, Ceram Tape and PVC, have similar characteristics in terms of flexibility.

### 3.5. Temperature Exposure

In order to test temperature stability, six identical chips were fabricated. These chips have a 3-layered structure (L1-PVC/L2-Ceram Tape/L3-PVC) where PVC layers have a thickness of 80 µm and lamination process was held at 150 °C. Chips were exposed for 5 min (in the preheated oven) to the temperatures from 80 °C to 180 °C, with the step of 20 °C. Figure 12a shows six chips before, whereas Figure 12b–d show chips after the temperature exposure. On the temperature of 100 °C PVC foil is heated, but there is no deformation of the channel after medium pressure to the foil. Moreover, at 120 °C chip still looks intact but there is a mild deformation of the channel while applying medium pressure. At 140 °C there is a visible deformation of chip, air bubbles are scattered through the surface and channel suffers significant deformation under medium pressure, but channels are still functional. Further, at 160 °C and higher PVC foil is melting and closes the channel. Taking into account that the proposed microfluidic chip is composed of two materials, the temperature stability of the chip is predominantly determined with the characteristics of PVC layer. Ceram Tape is “green” unfired tape which has good thermal characteristic and can be exposed to harsh environment [37]. Thermal expansion coefficient of PVC and Ceram Tape are 50 ppm/°C [40] and 5.5 ppm/°C [35], respectively. These indicates that PVC is more critical layer which restricts the temperature limit. Therefore, the proposed microfluidic chip can be exposed to the temperature up to 120 °C which is acceptable limitation for the majority of the real applications, including different isothermal amplification or cell incubation processes. 

### 3.6. Microfluidic Testing

To test adhesion and leaks of the chip with fluid inside, a simple microfluidic experimental set-up was used. This set-up consisted of a syringe pump, syringes, tubes and connectors as well as chip holder with patches and the proposed PVC/Ceram Tape/PVC chip, Figure 13.

For determination of maximal flow rate through the chips, two sets of six chips were made. For the first set, chips with simple channel design cut in Ceram Tape were laminated with 80 µm PVC foil on both sides on the temperature from 130 °C to 180 °C with the step of 10 °C. While, for the second test, 125 µm PVC foils were used in the same temperature range. Flow rate was tested in five ranges described in detail in Table 2, for 20 s each. All chips resisted all flow rates in all ranges without any leakages. At flow rate of 15 mL/min silicon patches that holds the tubes started to leak, so conducting tests on higher flow rates were not possible. For comparison, two other sets of chips were made with only one change—instead of Ceram Tape layer we used 80 µm PVC foil for the first set and 125 µm PVC foil for the second test as middle layer cut on the plotter cutter as in standard xurography technique with the same design used for Ceram Tape. Only two of the chips have been successful with unobstructed channels or inlets/outlets and resisted flow rate up to 5 µL/min without leakages. These tests demonstrated that presented technology for fabrication microfluidic chips can resist 3000 times higher flow rates than chips manufactured using standard xurography technique.

## 4. Application Examples

To demonstrate a potential of the proposed technology, two microfluidic chips have been fabricated and tested: 3D multilayered microfluidic chip for particle filtration and microfluidic microwave sensors. 

The 3D multilayered chip that comprises five layers, Figure 14a, was fabricated using a combination of laser micromachining and PVC lamination processes. The Layers 2–4 were realized using the Ceram Tapes, while top and bottom layers, i.e., Layer 1 and Layer 5 were realized using PVC foils. The laser micromachining process has been used for the preparation of tapes and cutting of microfluidic channel, mixer, serpentine and filter. The width of the microfluidic channels was set to 200 µm, while interconnection holes between layers and holes for the inlet and outlet were realized with diameter of 2 mm. In the next step, the Ceram Tapes were stacked, and laminated using Uniaxial press Carver 3895CEB under the pressure of 2200 kg and temperature of 80 °C, for 3 min. The precise alignment between Ceram Tape layers was obtained using a mould with fixed pins. Ceram Tape layers, previously cut with the laser together with holes for alignments, were stuck in the mould before the lamination process. The Whatman^®^ 1441–150 filter paper with grade of 20–25 µm previously cut with laser has been embedded in the Layer 3 during stacking of the layers before lamination. To avoid leakage, the size of the filter is enlarged to 5 mm, whilst the filtration area has been realized with the radius of 2 mm. The filter is integrated between two layers in the sandwich structure. The cutting plotter was used to engrave inlets, outlet and edges in the plastic laminating films used for the top and bottom layers. The top and bottom enclosure layers were assembled in the final step using lamination process. The layout of the fabricated 3D microfluidic chip is shown in Figure 14b.

To test functionality of the chip with filter unit, we made a custom solution of distillate water and pollen particles (*typha* pollen particles). As an experimental set-up, we used the same as for the determination of the maximal flow rate (shown in Figure 13). On two inlets water/pollen mixture was connected with 100 µL/min flow rate and output was kept in the 2.5 mL PMMA cuvette. Figure 15 shows the result of the filtration, the left cuvette contains water/pollen solution, the middle one is pure deionised water (used for comparison) and the right one is filtrated liquid. It can be seen that the quantity of pollen particles was significantly reduced. In detail, since *typha* pollen particles have a size of 22.7 ± 3 µm [41], and filter has grade of 20–25 µm, all medium and big particles were filtered; only some smaller ones passed through the filter.

Another example presents the resonant microwave microfluidic sensor presented in Figure 16. The sensor is realized using microstrip line topology in combination with the proposed chip. Realized sensor allows monitoring of the fluid properties flowing in the microchannel embedded between the microstrip line and ground plane, realized on top and bottom side of the chip, respectively. The top layer consists of microstrip line with width of 2 mm and the ring resonators whose dimensions were marked in Figure 16. The conductive parts from the top and bottom side were realized using 40 µm thick conductive aluminum sticky tape precisely cut with the laser and accurately positioned on the microfluidic chip. The final layout of the proposed sensors with mounted end-launch SMA connectors (SMA Southwest Microwave 292-04A-5) used for the characterization is shown in Figure 16a. The microfluidic channel is cut in the Ceram Tape layer and positioned bellow the resonator, the most sensitive location in the proposed design. The sensor operating principle is based on the resonant shift measurement in the reflection characteristic. The change of the fluid properties in the microfluidic channel causes the change of effective dielectric constant of the substrate, which results in the change of effective permittivity of the microstrip. Consequently, the resonant frequency of the ring resonators changes, which affect the reflection characteristic. The sensor response is measured in the frequency range between 1 GHz and 8 GHz using two ports Agilent 8501C Vector Network Analyser. The measurement results for different fluid placed in the microfluidic channel are shown in Figure 17a. The sensitivity of the proposed sensor, defined as an influence of the dielectric constant of the fluid in the microfluidic channel to the frequency shift is shown in Figure 17b. The measurement results show that the change of the fluid permittivity from 1 to 80.1 causes the frequency shift of 1.37 GHz. Therefore, the proposed sensor is characterized with good sensitivity, especially for the low values of dielectric constant.

## 5. Discussion

In this paper, we proposed one solution for microfluidic chip fabrication that can be easily used for the rapid fabrication of robust microfluidic chips. The proposed fabrication process combines PVC foil and Ceram Tape, and relies on cost-effective lamination technique and laser micromachining process. We demonstrate that small microfluidic channels with sharp channel edges and small surface deformation can be easily and precisely cut using a laser. An advantage of the proposed technology is also the possibility to create a multilayered chip which is mechanically flexible. The multilayered configuration can be formed using simple lamination process, where each layer can be made and tested separately, before lamination. Therefore, high reliability and reproducibility can be achieved, and the complex 3D geometry can be realized. Prior to lamination, the screen or ink-jet printing process can be used to print electrodes or sensitive layer directly on foil or Ceram Tape (green tape in unfired state). In that manner, conductive or some other materials (nanomaterials or biomaterials) can be directly printed on the tape or foil before the lamination, which is not the case in many other technologies reported so far. Additionally, it is worth mentioning that proposed hybrid technology does not require lithography process, fabrication of any support layer or mould. The manufacturing process is very fast and fabrication time needed to produce one microfluidic chip using described technology in laboratory conditions is less than a minute, while some other technologies (such as PDMS or LTCC) requires a couple of hours. 

In the Table 3 we compare different microfluidic fabrication technologies in terms of optical and mechanic characteristics, temperature exposure, fabrication time and complexity, integration, and bio-compatibility. In addition, the advantages and disadvantages of different microfluidic technologies are summarized in the Table 4.

The proposed microfluidic structures have been manufactured combining mechanically flexible PVC foils and Ceram Tapes, using the following equipment: cuter, laser (for micromachining) and laminator to create the compact, robust chip. The proposed process does not require a costly clean room facility and lithographic process. Therefore, the lower starting investment potentially could be only the 3D printing process, but using the 3D printing method it is not possible to manufacture mechanically flexible microfluidic chips with good optical characteristics, such as those presented in this paper.

Optical characterization shows that microfluidic chips fabricated using proposed technology have a slightly lower percentage of optical transmittance comparing with PDMS, but greater than all transparent 3D printed materials. However, the presented hybrid technique in this paper that combines PVC foils and green (unfired) flexible tapes can be used to create complex multi-layered geometries. On the other hand, proposed technology is much simpler in terms of the manufacturing complexity, the cost of manufacturing is lower, and time needed for manufacturing of single microfluidic chip used proposed technology is significantly shorter than the time required for the production of PDMS or LTCC chips. Therefore, we believe that the presented hybrid technology shows a good potential for the realization of the novel class of lab-on-chip devices with high reliability and reproducibility that can find their fields of application in the realization of robust, disposable microfluidic chips for rapid in-field testing and can be easily integrate with different sensors and electronics. 

It can be noted that PVC and Green Tape as materials limit potential field of applications. However, PVC is used to transmit and store fluid in many medical devices, such as urine, saliva, blood, and blood products without any changes in material composition. Our further research in this field will be focused on the investigation and analysis of the bio-compatibility and bio-applicability of the proposed microfluidic chips. One of the main advantages of proposed microfluidic chip technology are the utilization of low-cost materials, so that the chips can be easily disposable after single use. Further, there is wide range of microfluidic experiments which includes a short time of medium retention in the PVC chip. Therefore, we believe that the proposed technology shows good potential for the realization of the novel disposable microfluidic chip for rapid in-field testing and that they can find their fields of applications in medical analysis, water analysis, and control of food quality.

## 6. Conclusions

In this paper, we propose a novel technology concept that can be easily used for the rapid fabrication of low-cost microfluidic chips. The proposed fabrication process combines PVC foils and Ceram Tapes, and relies on the cost-effective xurography technique and laser micromachining process. The comprehensive study of dielectric, optical, structural, mechanical, and temperature properties of fabricated microfluidic chips and used materials have been performed. The proposed chips show good optical, mechanical, and thermal characteristics and excellent resistance to high flow rates. An important advantage of the proposed technology is also the possibility to create and test every layer separately before lamination. Therefore, high reliability and reproducibility can be achieved. Prior to lamination, the screen or inject printing process can be used to print electrodes directly on foil or Ceram Tape. In that manner, complex multilayered microfluidic chips can be fabricated using combinations of laser micromachining process for creation of complex microchannel geometries, screen/ink-jet printing process for deposition of conductive paste or biomaterial directly on tape or foil, and lamination processes for bonding. Several microfluidic chips, including 3D multilayered microfluidic chip for particle filtration and microfluidic microwave sensors have been realized and tested. The proposed results show a good potential of the proposed technology concept for the realization of the novel class of lab-on-chip devices with high reliability and reproducibility or realization of low-cost disposable microfluidic chips.

## Figures and Tables

**Figure 1 sensors-19-01719-f001:**
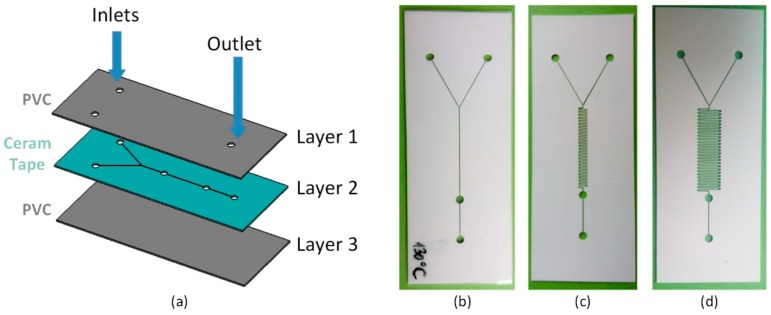
Microfluidic chips fabricated using proposed hybrid technology (**a**) 3D model of the microfluidic chip (Layer1-PVC, Layer2-Ceram Tape, and Layer3-PVC) (**b**) simple channel, (**c**) short serpentine, and (**d**) long serpentine.

**Figure 2 sensors-19-01719-f002:**
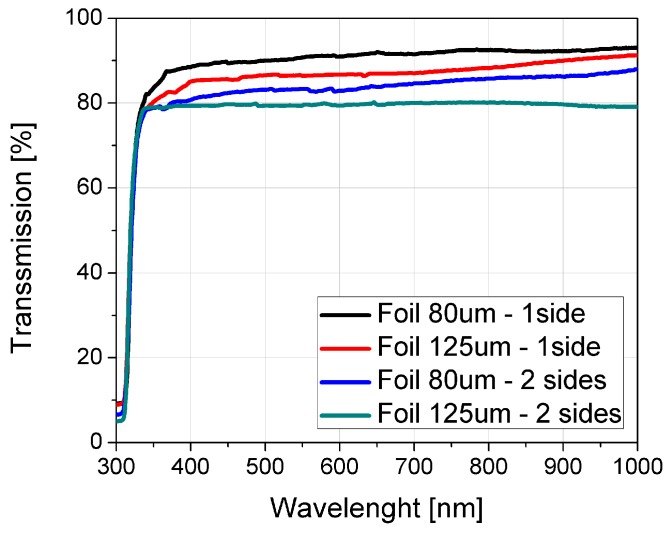
Optical transmission as a function of wavelength for four different configurations.

**Figure 3 sensors-19-01719-f003:**
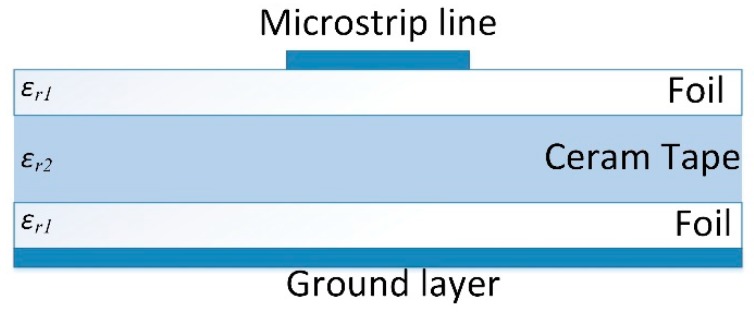
The configuration of the microstrip line on multilayered substrate used for permittivity characterization.

**Figure 4 sensors-19-01719-f004:**
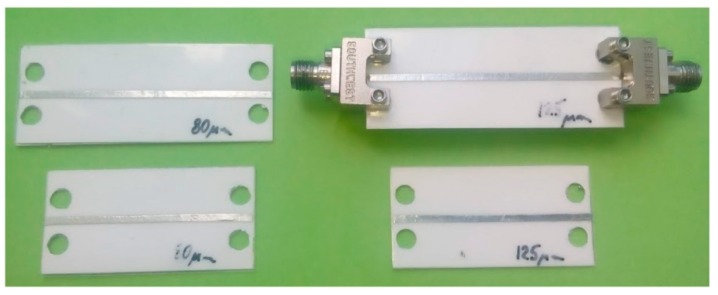
The layouts of the fabricated microstrip lines with SMA connectors used for permittivity characterization.

**Figure 5 sensors-19-01719-f005:**
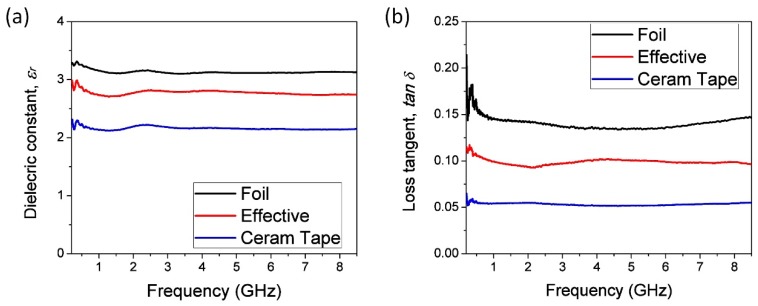
Extracted value of the: (**a**) effective permittivity, and (**b**) tan *δ*.

**Figure 6 sensors-19-01719-f006:**
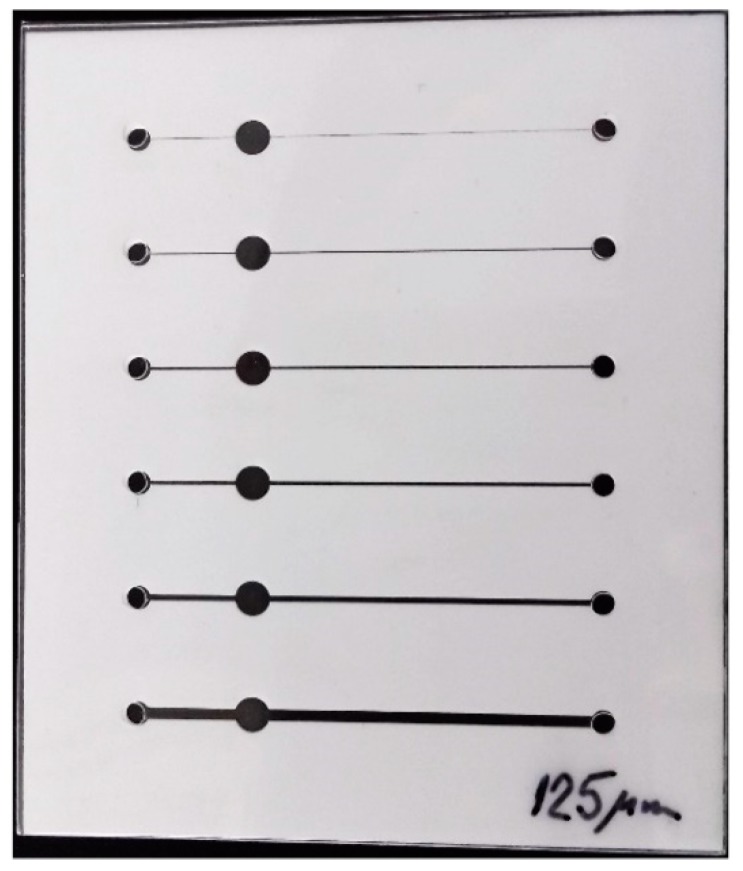
Photographs of chips with different widths of the channel.

**Figure 7 sensors-19-01719-f007:**
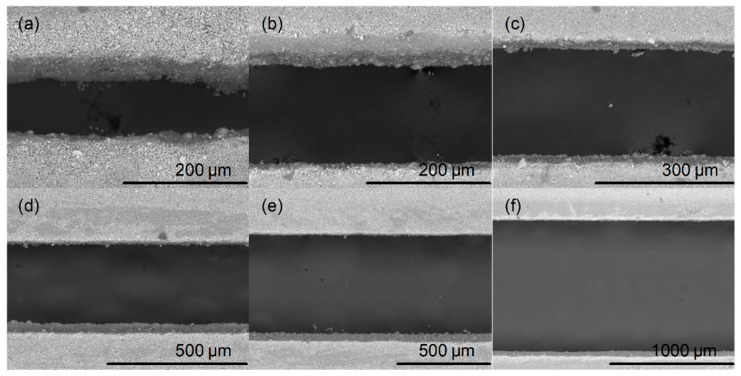
SEM micrographs of laser cut channels with different widths at optimal magnifications: (**a**) 50 μm, ×400, (**b**) 100 μm, ×400, (**c**) 200 μm, ×300, (**d**) 300 μm, ×180, (**e**) 500 μm, ×120, (**f**) 1 mm, ×80.

**Figure 8 sensors-19-01719-f008:**
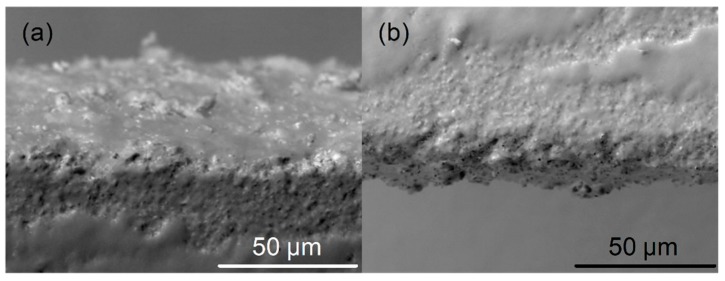
SEM micrographs of the laser cut channels edges on x1k2 magnification: (**a**) left edge, and (**b**) right edge.

**Figure 9 sensors-19-01719-f009:**
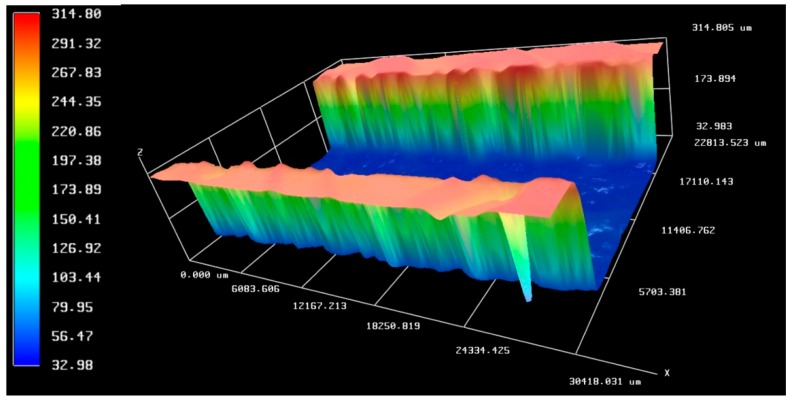
Profile thickness and roughness of the channel cut in middle layer (Ceram Tape layer).

**Figure 10 sensors-19-01719-f010:**
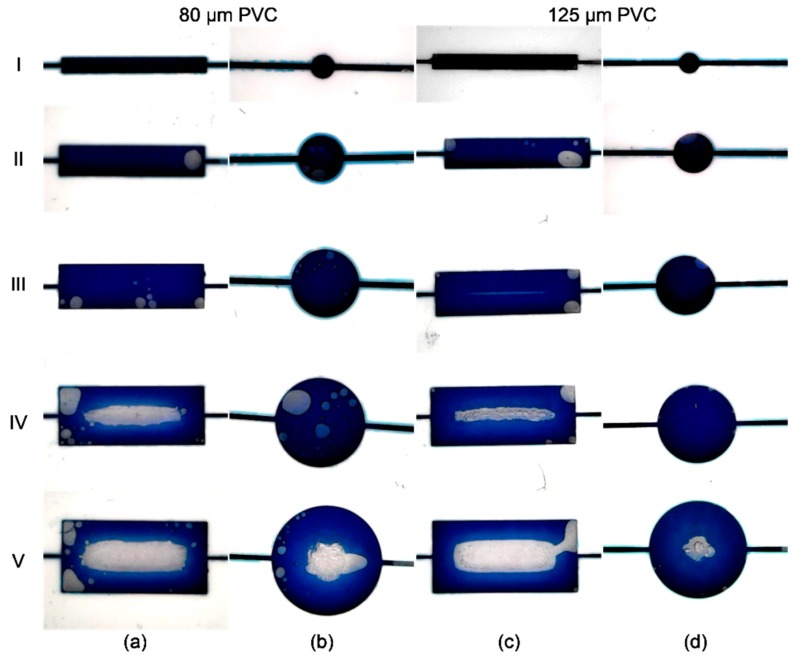
Tests of limitations for the microfluidic chambers with different widths/diameters: (**a**) rectangles laminated with 80 μm PVC foil; (**b**) circles laminated with 80 μm PVC foil; (**c**) rectangles laminated with 125 μm PVC foil; and (**d**) circles laminated with 125 μm PVC foil.

**Figure 11 sensors-19-01719-f011:**
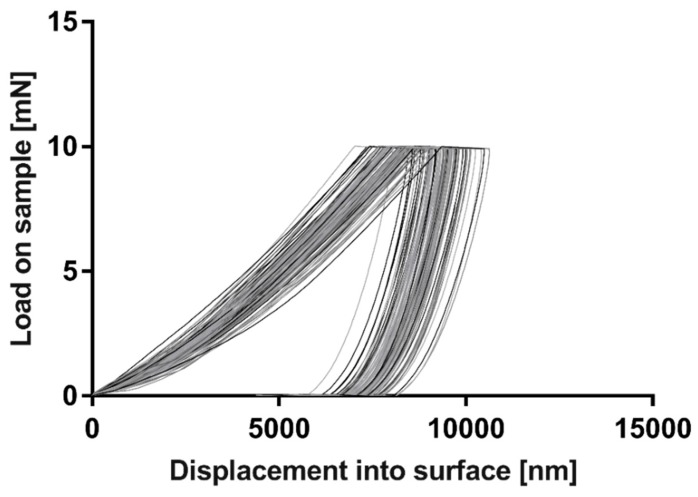
Load vs. Displacement curves for non-baked Ceram Tape.

**Figure 12 sensors-19-01719-f012:**
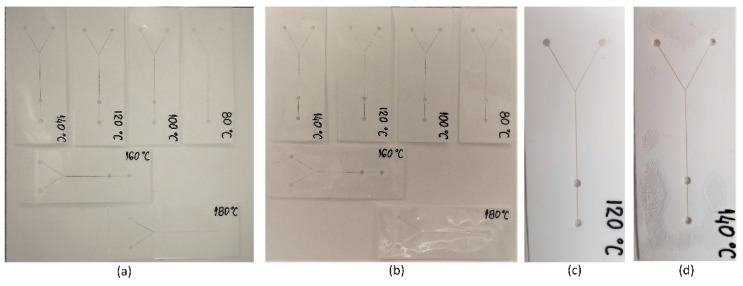
Thermally treated chips: (**a**) before temperature exposure, (**b**) after temperature exposure, (**c**) chip at 120 °C after exposure, without visible deformation on the chip, (**d**) chip at 140 °C with channel deformation and air bubbles on the chip.

**Figure 13 sensors-19-01719-f013:**
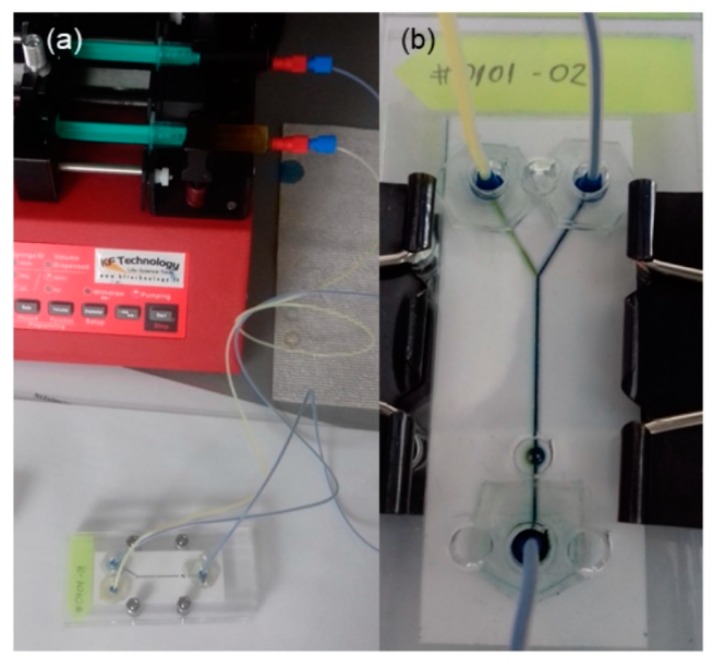
Microfluidic experimental set-up: (**a**) Syringe pump, tubes connectors and chip holder, and (**b**) Chip holder with mounted microfluidic chip.

**Figure 14 sensors-19-01719-f014:**
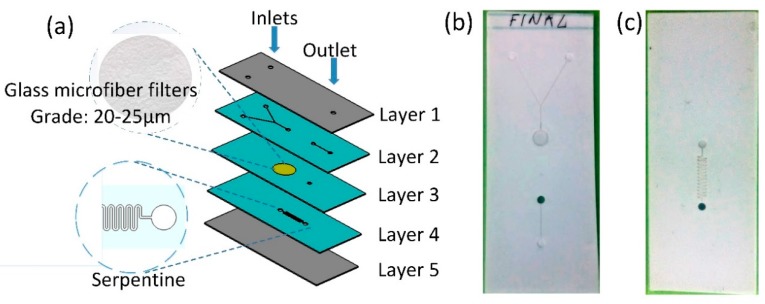
(**a**) Exploded view of multi-layered 3D microfluidic chip, (**b**) Top and (**c**) Bottom layer of the fabricated microfluidic chip.

**Figure 15 sensors-19-01719-f015:**
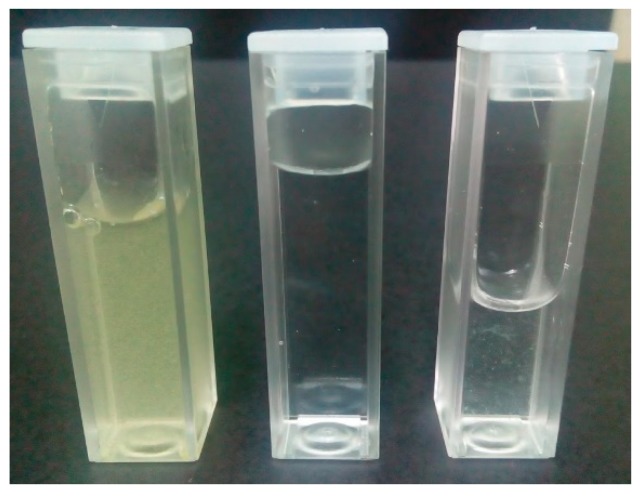
Filtration results of the microfluidic chip with filtration unit. Left cuvette contains water/pollen solution, middle one is pure deionised water (used for comparison) and right one is filtrated liquid.

**Figure 16 sensors-19-01719-f016:**
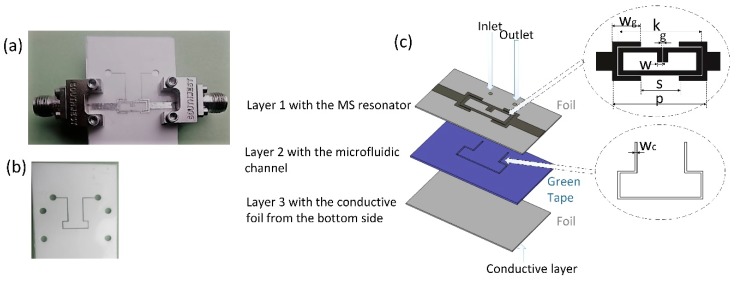
Layout of the resonant microwave microfluidic sensor: (**a**) fabricated circuit with SMA connectors, (**b**) microfluidic chip, and (**c**) layout of the proposed microfluidic microwave sensor. *w_g_* = 2.9 mm, *k* = 8.5 mm, *g* = 0.1 mm, *w* = 0.5 mm, *s* = 3.9 mm, *p* = 9.7 mm and *w_c_* = 0.2 mm.

**Figure 17 sensors-19-01719-f017:**
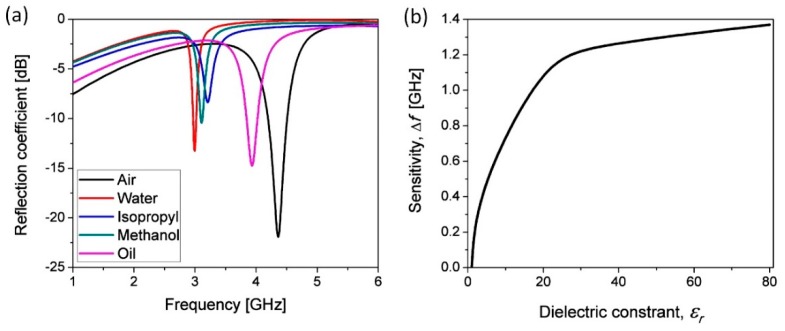
Measured results of the proposed sensor with different fluids inside the microfluidic channel: (**a**) Reflection characteristic; and (**b**) Sensitivity of the proposed microwave sensor.

**Table 1 sensors-19-01719-t001:** Exact fabrication parameters used for microfluidic chip fabrication.

**Step 1: Laser cutting**		
**Parameter**	**Value**	**Unit**
**Current**	28	mA
**Frequency**	10	kHz
**Speed**	15	mm/s
**Step 2: Uniaxial press**		
**Parameter**	**Value**	**Unit**
**Pressure**	2200	kg
**Temperature**	80	°C
**Time**	3	min
**Step 3: Plotter cutting**		
**Parameter**	**Value**	**Note**
**Cutting speed—inlet/outlet**	30 cm/s	from range 1–60
**Cutting speed—border**	60 cm/s	from range 1–60
**Cutting speed—channel**	10 cm/s	from range 1–60
**Cutting force—80 µm PVC**	19	from range 1–38
**Cutting force—125 µm PVC**	26	from range 1–38
**Step 4: Lamination**		
**Parameter**	**Value**	**Note**
**Temperature**	120–180 °C	150 °C if not indicated otherwise
**Speed**	1 cm/min	from range 1–9

**Table 2 sensors-19-01719-t002:** Flow rate ranges and steps used in flow rate testing.

Range	From	To	Step
**I**	0.1 µL/min	1.0 µL/min	0.1 µL/min
**II**	1 µL/min	10 µL/min	1 µL/min
**III**	10 µL/min	100 µL/min	10 µL/min
**IV**	100 µL/min	1 mL/min	100 µL/min
**V**	1 mL/min	15 mL/min	1 mL/min

**Table 3 sensors-19-01719-t003:** Characteristics and advantages/disadvantages of different microfluidic fabrication technologies.

Technology	Proposed	PDMS	LTCC	3D Printing	Xurography
**Optical transparency**	good	excellent	none can be achieved by bonding of LTCC with other transparent materials	poor	good
**Mechanical flexibility**	flexible	flexible and stretchable	none	none	flexible
**Channel edge roughness**	excellent	good	good	poor	poor
**Bio-compatibility**	good	excellent	excellent	good	good
**Temperature exposure**	up to 120 °C	up to 200 °C properties of PDMS changes with temperature	very high >1000 °C	up to 160 °C	up to 120 °C
**Possibility to crate complex geometries**	excellent	good but requires additional mould and supporting materials	excellent	excellent but requires supporting layer	medium
**Fabrication time** **simple/complex geometry**	1–10 min	Couple of hours	>6 h	10–180 min	1–10 min
**Fabrication complexity**	simple/medium	complex	complex	simple	simple

**Table 4 sensors-19-01719-t004:** The advantages and disadvantages of different microfluidic technologies.

Technology	Advantages	Disadvantages
**Proposed**	fabricate and test every layer separately 3000 times higher flow rates than xurography chips the screen or inkjet printing process can be used to print electrodes or sensitive layer directly on foil or Ceram Tape low cost chips rapid fabrication	limited time usage of the chip
**PDMS**	long lasting chips	requires non-trivial lithography method manufacturing of complex 3D channels is really challenging task high cost chips
**LTCC**	fabricate and test every layer separately the screen printing process can be used to print electrodes directly on green tape high operating temperature	changing shape and dimensions of the channels during lamination or firing process optical transparency
**3D printing**	low cost chips	low channel resolution leak-proof problem
**Xurography**	fabricate and test every layer separately low cost chips the screen or inkjet printing process can be used to print electrodes or sensitive layer directly on foil	uneven edges of the microchannels limited time usage of the chip manufacturing of complex 2D and 3D channels is really challenging task

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
