# Peer review of "Novel Cost-Effective Microfluidic Chip Based on Hybrid Fabrication and Its Comprehensive Characterization"

_sensors, 2019, doi:10.3390/s19071719_

Reviewer 1 Report

This paper reports a lithography method for fabricating hybrid microfluidic devices using the combination of laser micromachining and xurographic method. The as-fabricated devices were used in particle filtration and microwave sensor.

The paper is not acceptable for publication for the following reasons:

(1) There have been many works on using xurographic method to fabricate microfluidic devices. The authors need to point out the advantages of the current method---my specific concern is that the usage of PVC would place limit on the applications in terms of heat stability and organic solvents compatibility.

(2) In the introduction, more information of the Ceram Tape should be provided.

(3) The writing of the manuscript needs substantial improvement. There are many grammar mistakes. Many descriptions and discussions are not clear.

Author Response

Dear Sir/Madam,

We appreciate the careful reviewing of our manuscript and we gratefully acknowledge the reviewers and editor for their useful comments. We have modified the manuscript to answer the questions raised by the reviewers. Our answers, some additional comments and explanations are given in the text below and the changes are made in the resubmitted version of our manuscript.

We thank you for the opportunity to improve our paper, including your comments and suggestions.

Sincerely,

The authors

Reply to the Reviewer’s Comments

Answers to the First Reviewer

Reviewer’s comment 1: There have been many works on using xurographic method to fabricate microfluidic devices. The authors need to point out the advantages of the current method---my specific concern is that the usage of PVC would place limit on the applications in terms of heat stability and organic solvents compatibility.

Author’s response 1: In order to respond to this valuable comment, the new paragraph has been added in the revised version of this manuscript, which emphasizes the advantages of the proposed method:

Section 5, Page 13, lines 342 - 358 “In this paper, we proposed one solution for microfluidic chip fabrication that can be easily used for the rapid fabrication of robust microfluidic chips. The proposed fabrication process combines PVC foil and Ceram Tape, and relies on cost-effective lamination technique and laser micromachining process. We demonstrate that small microfluidic channels with sharp channel edges and small surface deformation can be easily and precisely cut using laser. Advantage of the proposed technology is also possibility to create a multilayered chip which is mechanically flexible. The multilayered configuration can be formed using simple lamination process, where each layer can be made and tested separately, before lamination. Therefore, high reliability and reproducibility can be achieved and the complex 3D geometry can be realized. Prior to lamination, the screen or inkjet printing process can be used to print electrodes or sensitive layer directly on foil or Ceram Tape (green tape in unfired state). In that manner, conductive or some other materials (nanomaterials or biomaterials) can be directly printed on tape or foil before the lamination, which is not case in many other technologies reported so far. Additionally, it is worth mentioning that proposed hybrid technology does not require lithography process, fabrication of any support layer or mould. The manufacturing process is very fast and fabrication time needed to produce one microfluidic chip using described technology in laboratory conditions is less than a minute, while some other technologies (such as PDMS or LTCC) requires a couple of hours.”

Section 5, Page 13, lines 390 - 382 “Therefore, we believe that the presented hybrid technology shows a good potential for the realization of the novel class of lab-on-chip devices with high reliability and reproducibility that can find their fields of application in the realization of robust, disposable microfluidic chips for rapid in-field testing and can be easily integrate with different sensors and electronics.”

In section 3.5. Temperature exposure we also included to following sentences, which explain heat issue:

Section 3.5, Page 10, lines 242- 248:“Taking into account that the proposed microfluidic chip is composed of two materials, the temperature stability of the chip is predominantly determined with the characteristics of PVC layer. Ceram Tape is “green” unfired tape which has good thermal characteristic and can be exposed to harsh environment [*1]. Therefore, the proposed microfluidic chip can be exposed to the temperature up to 100°C which is acceptable limitation for the majority of the real applications, including different isothermal amplification or cell incubation processes”

Additional reference:

[*1]   M. Franz, I. Atassi, A. Maric, B. Balluch, M. Weilguni, W. Smetana, C.P. Kluge, G. Radosavljevic, “Material Characteristics of the LTCC Base Material CeramTape GC”, 35th Int. Spring Seminar on Electronics Technology, pp. 276-281, 2012.

The bio-compatibility of the used materials are comment in more details in introduction section and discussion. The following sentences has been added to the manuscript: 

Section: 1, Page 2, lines 79- 93: “The proposed chip combines two materials, PVC and Green Tape, both with relatively good bio-compatible characteristics. PVC is a widely used thermoplastic material in the medical device industry and it is dominantly used for the storage of fluids, dialysis solutions, blood, and blood products [*2-*3]. PVC is capable to accept or transmit a variety of fluid without any significant changes in composition or properties [*4]. It is characterized by good biocompatibility, which can be further increased by appropriate surface modification [*5].

Ceram Tape is a LTCC glass ceramic base material composed of an anorthite glass (calcium aluminosilicate) with ceramic filler. This green tape is appropriate for manufacturing fine structures for electronic and microfluidic application intended to work in harsh environment. The bio-compatibility of Ceram Tape GC for cell grown has been confirmed in [*1]. To investigate the bio–compatibility, proliferation, viability, and adherence of cells on Ceram Tape GC, HeLa (human, cervix epithelial) cells have been grown on sintered tapes using standard test procedures, and the results confirm that Ceram Tape GC material does not affect the viability and adherence of the cells. On the other hand, a number of short or long time exposure of the microfluidic chip to different fluids have been accomplished to approve bio-compatibility of the Ceram Tape GC material [*6-*7].”

Additional references:

[*2]   Zhao, X. Update on Medical Plasticised PVC, 1st ed.; Smithers Rapra Press: Shrewsbury, Shropshire, United Kingdom, 2009.

[*3]   Sastri, V.R. Plastics in Medical Devices: Properties, Requirements and Applications, 1st ed. Elsevier, Burlington, USA, 2010.

[*4]   Staff P.D.L. Chemical Resistance, Volume 1: Thermoplastics, 2nd ed., William Andrew, NY, USA, 1994.

[*5]   Yianni J.P. Making PVC more biocompatible. Med Device Technol. 1995, 6(7): 20-6, 28-9.

[*6]   Malecha, K., Remiszewska, E., Pijanowska D.G. Technology and application of the LTCC-based microfluidic module for urea determination, Microelectr. International, 2015, 32(3):126-132

[*7]   Smetana, W., Balluch, B., Atassi, I., Gvichiyarahim K.E., Gaubitzer, E., Edetsberger, M., Koehler, G. A Biological Monitoring Module based on a Ceramic Microfluidic Platform, Proceedings of the Int. Conf. on Biomedical Electronics and Devices, Porto, Portugal, 2009, 14-17 January, 2009.

And the following paragraph is added to the Section 5. Discussion:

Section 5, Page 13, lines 383 - 392 It can be noted that PVC and Green Tape as materials limit potential field of applications. However, PVC is used for transmit and storage fluid in many medical devices, such as urine, saliva, blood, and blood products without any significant changes in material composition. Our further research in this field will be focus on investigation and analysis of the bio-compatibility and bio-applicability of the proposed microfluidic chips. One of the main advantages of proposed microfluidic chip technology are utilization of low cost materials, so that the chips can be easily disposable after single use. Further, there is wide range of microfluidic experiments which includes short time of medium retention in the PVC chip. Therefore, we believe that the proposed technology shows a good potential for the realization of the novel disposable microfluidic chip for rapid in-filed testing and that they can find their fields of applications in medicinal analysis, water analysis, and control of food quality.”

Two addition tables, Table 3 and Table 4 have been added in the revised manuscript. In the Table 4 we compare the advantages and disadvantages of different microfluidic fabrication technologies, while Table 3 we compare different microfluidic fabrication technologies in term of optical and mechanic characteristics, temperature exposure, fabrication time and complexity, integration, and bio-compatibility.

Reviewer’s comment 2: In the introduction, more information of the Ceram Tape should be provided.

Author’s response 2: We totally agree with this comment. In order to cover this comment, the following sentences have been incorporated in the revised version of the manuscript:

Section 1, Page 2, lines 85-87: “CeramTape is a LTCC glass ceramic base material composed of an anorthite glass (calcium aluminosilicate) with ceramic filler. This green tape is appropriate for manufacturing fine structures for electronic and microfluidic application intended to work in harsh environment.”

Reviewer’s comment 3: The writing of the manuscript needs substantial improvement. There are many grammar mistakes. Many descriptions and discussions are not clear.

Author’s response 3: We appreciate this comment. We believe that revised manuscript is better in this context, bearing in mind that we have performed carefully proof-re

Reviewer 2 Report

Review comments

“Novel cost-effective microfluidic chip based on hybrid fabrication and its comprehensive characterization” introduces the use of new microfluidic materials and applications in this manuscript. However, I have a skeptical idea about the practical application of micro levels in spite of authors’ real applications. As a result, I acknowledge the originality of the paper, but I feel sorry that there is a limit to the use of this manuscript by readers. It would be nice to fix the following parts.

1) Grammatical errors: Many errors have found through the manuscript. A full revision is required.

2) Cost-effective: I personally think it takes expensive equipment to make patterns in PVC. What do you think?

3) Clogging of channels: The fact that there is no need for a clean room is a big advantage, but you have to worry about the possibility of clogging problems caused by external dust.

4) Optical transmittance: Your result shown in Fig. 2 is better than 80%. As of today, PDMS is more than 90%. Why should we use PVC instead of PDMS? I think PDMS is a better selection to make microfluidic chips. In this case, you must mention the disadvantage of PDMS in the introduction parts.

5) Alignment: It is necessary to mention how to make a chip alignment even though you suggest in Figure 14. If the pattern of the microfluidic chip is complex, the method mentioned in this paper is unlikely to be used.

6) Applications: You have tested the possibility of actual applications, but you have not done the bio experiment which is the most use of the microfluidic chip as of today. The bio applicability of PVC needs to be reinforced. It would be nice if the experiment was added, but if not, it would be nice to mention it in the introduction or the results section.

7) Discussion: I encourage you to combine Discussion and Conclusions into Conclusions.

Author Response

Dear Sir/Madam,

We appreciate the careful reviewing of our manuscript and we gratefully acknowledge the reviewers and editor for their useful comments. We have modified the manuscript to answer the questions raised by the reviewers. Our answers, some additional comments and explanations are given in the text below and the changes are made in the resubmitted version of our manuscript.

We thank you for the opportunity to improve our paper, including your comments and suggestions.

Sincerely,

The authors

Reply to the Reviewer’s Comments

Answers to the Second Reviewer

Reviewer 2:

 “Novel cost-effective microfluidic chip based on hybrid fabrication and its comprehensive characterization” introduces the use of new microfluidic materials and applications in this manuscript. However, I have a skeptical idea about the practical application of micro levels in spite of authors’ real applications. As a result, I acknowledge the originality of the paper, but I feel sorry that there is a limit to the use of this manuscript by readers. It would be nice to fix the following parts.

Reviewer’s comment 1: Grammatical errors: Many errors have found through the manuscript. A full revision is required.

Author’s response 1: Thank you for this comment. As the respectable reviewer suggested, we performed a full revision of the manuscript in the context writing style and grammatical errors and we believe that revised manuscript is better from this point of view.

Reviewer’s comment 2: Cost-effective: I personally think it takes expensive equipment to make patterns in PVC. What do you think?

Author’s response 2: In order to cover this comment the following paragraphs have been included in the revised version of the manuscript

Section 5, Page 15, Lines 365 – 370: “The proposed microfluidic structures have been manufactured combining mechanically flexible PVC foils and Ceram Tapes, using the following equipment: cuter, laser (for micromachining) and laminator to create compact, robust chip. The proposed process does not require costly clean room facility and lithographic process. Therefore, the lower starting investment potentially could be only 3D printing process, but using 3D printing method it is not possible to manufacture mechanically flexible microfluidic chips with good optical characteristics, such as those presented in this paper.”

Section 5, Page 14, lines 38 – 392: “One of the main advantages of proposed microfluidic chip technology are utilization of low cost materials, so that the chips can be easily disposable after single use. Therefore, we believe that the proposed technology shows a good potential for the realization of the novel disposable microfluidic chip for rapid in-filed testing and that they can find their fields of applications in medicinal analysis, water analysis, and control of food quality.”

Reviewer’s comment 3: Clogging of channels: The fact that there is no need for a clean room is a big advantage, but you have to worry about the possibility of clogging problems caused by external dust.

Author’s response 3: Thank you for this comment. We are fully aware of this issue. Even the presented microfluidic chips are not manufactured in the clean room conditions, we have performed all other measures to fabricate and test chips in appropriate conditions such as: control the temperature and pressure in the laboratory, working with appropriate cloth and gloves, using part of the sticky carpet which exactly attract all dusty particles, etc. Taking all of this into account, we have not faced any problems with clogging of microfluidic channels with external dust. However, the completed process of proposed chip fabrication can be performed in clean room.

Reviewer’s comment 4: Optical transmittance: Your result shown in Fig. 2 is better than 80%. As of today, PDMS is more than 90%. Why should we use PVC instead of PDMS? I think PDMS is a better selection to make microfluidic chips. In this case, you must mention the disadvantage of PDMS in the introduction parts.

Author’s response 4: Authors really respect this comment. In order to cover this comment some additional sentences have been added to the Introduction and Discussion sections. It can be mention that microfluidic chips fabricated using PDMS can have slightly higher percentage of optical transmittance comparing with PVC foils. Nevertheless, PVC foil has better transparency comparing with all materials obtained using 3D printing process. However, presented hybrid technique in this paper that combine PVC foils and green (unfired) flexible tapes can be used to create complex multi-layered geometries. On the other hand, proposed technology is much simpler in terms of the manufacturing complexity, the cost of manufacturing is lower, and time needed for manufacturing of single microfluidic chip using proposed technology is significantly shorter than the time required for the production of PDMS chips. Therefore, we believe that proposed microfluidic chip can find their application as disposable chip for in-filed testing in combination with different optical sensors.

Two addition tables, Table 3 and Table 4 have been added in the revised manuscript. In the Table 4 we compare the advantages and disadvantages of different microfluidic fabrication technologies, while Table 3 we compare different microfluidic fabrication technologies in term of optical and mechanic characteristics, temperature exposure, fabrication time and complexity, integration, and bio-compatibility.

Reviewer’s comment 5: Alignment: It is necessary to mention how to make a chip alignment even though you suggest in Figure 14. If the pattern of the microfluidic chip is complex, the method mentioned in this paper is unlikely to be used.

Author’s Response 5: We really appreciate this comment. The precise alignment between Ceram Tape layers was obtained using a mould with fixed pins. For the lamination of the proposed multilayered chip we used mould with six fixed pins. These pins does not allow misalignment of any layers. Before the lamination different CeramTape layers, previously cut with the laser together with holes for alignments, were stuck in the mould, following with uniaxial press exposure. In that manner, the possibility of occurrence of misalignment is very low. Taking into account that the microfluidic channels are realized in the middle layer, and that the upper and lower PVC layers serve as channel lids, misalignment between PVC and Green Tape is not of great importance. In that case, misalignments only exist at the inlet and the outlet of the chip, and again it can be reduced using a specific holder that can reduce the misalignment during the lamination process. The following paragraph has been included in the revised version of the manuscript:

Section 4, Page X, lines 285 – 288: The precise alignment between Ceram Tape layers was obtained using a mould with fixed pins. Before the lamination different Ceram Tape layers, previously cut with the laser together with holes for alignments, were stuck in the mould.”

Reviewer’s comment 6: Applications: You have tested the possibility of actual applications, but you have not done the bio experiment which is the most use of the microfluidic chip as of today. The bio applicability of PVC needs to be reinforced. It would be nice if the experiment was added, but if not, it would be nice to mention it in the introduction or the results section.

Author’s Response 6: Authors indeed respect it comments because it is in line with our plans for the next steps in our research and studies. This, we kindly ask the reviewer to accept our explanation that we will perform bio experiment in the near future and present these results in some of our forthcoming papers. In this manuscript we include some explanation in Introduction and Disruption section regarding the bio-compatibility of used materials. The following sentence are added:

Section: 1, Page 2, lines 79- 93: “The proposed chip combines two materials, PVC and Green Tape, both with relatively good bio-compatible characteristics. PVC is a widely used thermoplastic material in the medical device industry and it is dominantly used for the storage of fluids, dialysis solutions, blood, and blood products [*2-*3]. PVC is capable to accept or transmit a variety of fluid without any significant changes in composition or properties [*4]. It is characterized by good biocompatibility, which can be further increased by appropriate surface modification [*5].

Ceram Tape is a LTCC glass ceramic base material composed of an anorthite glass (calcium aluminosilicate) with ceramic filler. This green tape is appropriate for manufacturing fine structures for electronic and microfluidic application intended to work in harsh environment. The bio-compatibility of Ceram Tape GC for cell grown has been confirmed in [*1]. To investigate the bio–compatibility, proliferation, viability, and adherence of cells on Ceram Tape GC, HeLa (human, cervix epithelial) cells have been grown on sintered tapes using standard test procedures, and the results confirm that Ceram Tape GC material does not affect the viability and adherence of the cells. On the other hand, a number of short or long time exposure of the microfluidic chip to different fluids have been accomplished to approve bio-compatibility of the Ceram Tape GC material [*6-*7].”

Additional references:

[*1]   M. Franz, I. Atassi, A. Maric, B. Balluch, M. Weilguni, W. Smetana, C.P. Kluge, G. Radosavljevic, “Material Characteristics of the LTCC Base Material CeramTape GC”, 35th Int. Spring Seminar on Electronics Technology, pp. 276-281, 2012.

[*2]   Zhao, X. Update on Medical Plasticised PVC, 1st ed.; Smithers Rapra Press: Shrewsbury, Shropshire, United Kingdom, 2009.

[*3]   Sastri, V.R. Plastics in Medical Devices: Properties, Requirements and Applications, 1st ed. Elsevier, Burlington, USA, 2010.

[*4]   Staff P.D.L. Chemical Resistance, Volume 1: Thermoplastics, 2nd ed., William Andrew, NY, USA, 1994.

[*5]   Yianni J.P. Making PVC more biocompatible. Med Device Technol. 1995, 6(7): 20-6, 28-9.

[*6]   Malecha, K., Remiszewska, E., Pijanowska D.G. Technology and application of the LTCC-based microfluidic module for urea determination, Microelectr. International, 2015, 32(3):126-132

[*7]   Smetana, W., Balluch, B., Atassi, I., Gvichiyarahim K.E., Gaubitzer, E., Edetsberger, M., Koehler, G. A Biological Monitoring Module based on a Ceramic Microfluidic Platform, Proceedings of the Int. Conf. on Biomedical Electronics and Devices, Porto, Portugal, 2009, 14-17 January, 2009.

And the following paragraph is added to the Section 5. Discussion:

Section 5, Page 15, lines 383 – 392: “It can be noted that PVC and Green Tape as materials limit potential field of applications. However, PVC is used for transmit and storage fluid in many medical devices, such as urine, saliva, blood, and blood products without any significant changes in material composition. Our further research in this field will be focus on investigation and analysis of the bio-compatibility and bio-applicability of the proposed microfluidic chips. One of the main advantages of proposed microfluidic chip technology are utilization of low cost materials, so that the chips can be easily disposable after single use. Further, there is wide range of microfluidic experiments which includes short time of medium retention in the PVC chip. Therefore, we believe that the proposed technology shows a good potential for the realization of the novel disposable microfluidic chip for rapid in-filed testing and that they can find their fields of applications in medicinal analysis, water analysis, and control of food quality.”

Reviewer’s comment 7: Discussion: I encourage you to combine Discussion and Conclusions into Conclusions.

Author’s response 7: According to the Reviewer’s valuable remarks, authors merge text from Discussion and Conclusions into one section (Conclusions in the resubmitted manuscript). However, we keep the section Discussion. In the revised manuscript new comments, compassion of the proposed technology with the existing ones, and some advantages and disadvantages of the proposed technologies are presented in section Discussion.

Reviewer 3 Report

Show a comparison chart with the previous works.

Author Response

Dear Sir/Madam,

We appreciate the careful reviewing of our manuscript and we gratefully acknowledge the reviewers and editor for their useful comments. We have modified the manuscript to answer the questions raised by the reviewers. Our answers, some additional comments and explanations are given in the text below and the changes are made in the resubmitted version of our manuscript.

We thank you for the opportunity to improve our paper, including your comments and suggestions.

Sincerely,

The authors

Reply to the Reviewer’s Comments

Answers to the Third Reviewer

Reviewer’s comment 1: Show a comparison chart with the previous works.

Author’s response 1: Two addition tables, Table 3 and Table 4 have been added in the revised manuscript. In the Table 4 we compare the advantages and disadvantages of different microfluidic fabrication technologies, while Table 3 we compare different microfluidic fabrication technologies in term of optical and mechanic characteristics, temperature exposure, fabrication time and complexity, integration, and bio-compatibility. In addition, section 5 was changed in the revised manuscript. New Section 5 compares the proposed technology with other recently used microfluidic technologies in details.

Table 3. Characteristics and advantages/disadvantages of different microfluidic fabrication technologies.

Technology

Proposed

PDMS

LTCC

3D   printing

Xurography

Optical   transparency

good

excellent

none

can be achieved by   bonding of LTCC with other transparent materials

poor

good

Mechanical   flexibility

flexible

flexible and stretchable

none

none

flexible

Channel   edge roughness

excellent

good

good

poor

poor

Bio-compatibility  

good

excellent

excellent

good

good

Temperature   exposure

up to 120 OC

up to 200 OC
  properties of PDMS changes with temperature

very high
  > 1000 OC

up to 160 OC

up to 120 OC

Possibility   to crate complex geometries

excellent

good

but requires   additional mould and supporting materials

excellent

excellent

but requires   supporting layer

medium

Fabrication   time

simple/complex   geometry

1-10 min

Couple of hours

> 6 hours

10 – 180 min

1-10 min

Fabrication   complexity

simple/medium

complex

complex

simple

simple

Table 4. The advantages and disadvantages of different microfluidic technologies.

Technology

Advantages

Disadvantages

Proposed

·     fabricate and test every layer separately

·     3000 times higher flow rates than   xurography chips

·     the screen or inkjet printing process can   be used to print electrodes or sensitive layer directly on foil or Ceram Tape

·     low cost chips

·     rapid fabrication

·  limited   time usage of the chip

PDMS

·     long lasting chips

·  requires   non-trivial lithography method

·  manufacturing of complex 3D channels is really   challenging task

·  high cost chips

LTCC

·     fabricate and test every layer separately

·     the screen printing process can be used to   print electrodes directly on green tape

·     high operating temperature

·  changing   shape and dimensions of the channels during lamination or firing process

·  optical   transparency

3D   printing

·     low cost chips

·  low   channel resolution

·  leak-proof   problem

Xurography

·     fabricate and test every layer separately

·     low cost chips

·     the screen or inkjet printing process can   be used to print electrodes or sensitive layer directly on foil

·  uneven   edges of the microchannels

·  limited   time usage of the chip

·  manufacturing   of complex 2D and 3D channels is really challenging task

Round  2

Reviewer 1 Report

The authors have addressed the issues raised by the reviewer, and I have the following minor questions:

1. What is the reason that the device can work only up to 100oC? What are the thermal expansion coefficients of PVC and the Ceram Tape?

2. I can still see many writing mistakes or confusing sentences in the manuscript. I listed a few examples below, and I suggest the authors to carefully go through the writing again or use a professional editor.

“capable to” (line 82)

“2 another sets” (line 266)

The sentence at line 287;

“end-lunch” (line 319).

Author Response

Dear Sir/Madam,

We appreciate the careful reviewing of our manuscript and we gratefully acknowledge the reviewers and editor for their useful comments. We have modified the manuscript to answer the questions raised by the reviewers. Our answers, some additional comments and explanations are given in the text below and the changes are made in the resubmitted version of our manuscript. We thank you for the opportunity to improve our paper, including your comments and suggestions. In the revised manuscript the English style is improved. We believe that revised manuscript is better in this context, bearing in mind that we have performed carefully proof-reading. If the reviewers or editor recommended additional English language editing, the MDPI English editing service will be used.

Sincerely,

The authors

Comment 1: The authors have addressed the issues raised by the reviewer, and I have the following minor questions: What is the reason that the device can work only up to 100oC? What are the thermal expansion coefficients of PVC and the Ceram Tape?

Author’s response 1: In order to respond to this valuable comment, the new sentences have been added in the revised version of this manuscript:

Taking into account that the proposed microfluidic chip is composed of two materials, the temperature stability of the chip is predominantly determined with the characteristics of PVC layer. Ceram Tape is “green” unfired tape which has good thermal characteristic and can be exposed to harsh environment [37]. Thermal expansion coefficient of PVC and Ceram Tape are 50 ppm/°C [40] and 5.5 ppm/°C [35], respectively. These indicates that PVC is more critical layer which restricts the temperature limit. Therefore, the proposed microfluidic chip can be exposed to the temperature up to 120°C which is acceptable limitation for the majority of the real applications, including different isothermal amplification or cell incubation processes.

Comment 2. I can still see many writing mistakes or confusing sentences in the manuscript. I listed a few examples below, and I suggest the authors to carefully go through the writing again or use a professional editor.

“capable to” (line 82)

“2 another sets” (line 266)

The sentence at line 287;

“end-lunch” (line 319).

Thank you for this comment. As the respectable reviewer suggested, we performed a full revision of the manuscript in the context writing style and grammatical errors and we believe that revised manuscript is better from this point of view. If the reviewers or editor recommended additional English language editing, the MDPI English editing service will be used.

Reviewer 2 Report

This manuscript still contains grammatical errors from the abstract section. Could you please check the entired manuscript ? I am very happy to read your recent revised manuscript.

Author Response

Dear Sir/Madam,

We appreciate the careful reviewing of our manuscript and we gratefully acknowledge the reviewers and editor for their useful comments. We have modified the manuscript to answer the questions raised by the reviewers. Our answers, some additional comments and explanations are given in the text below and the changes are made in the resubmitted version of our manuscript. We thank you for the opportunity to improve our paper, including your comments and suggestions. In the revised manuscript the English style is improved. We believe that revised manuscript is better in this context, bearing in mind that we have performed carefully proof-reading. If the reviewers or editor recommended additional English language editing, the MDPI English editing service will be used.

Sincerely,

The authors
